# Distribution of Human T-Lymphotropic Virus (HTLV) and Hepatitis C Co-infection in Bahia, Brazil

**Felicidade Mota Pereira**[1,2]**, Maria da Conceição Chagas de Almeida**[3]**, Fred Luciano Neves Santos**[1]**, Roberto Perez Carreiro**[4]**, Bernardo Galvão-Castro**[1,5]**, Maria Fernanda Rios Grassi**[1,5]***

1 Advanced Public Health Laboratory, Gonçalo Moniz Institute, FIOCRUZ-BA, Salvador, BA, Brazil,
2 Gonçalo Moniz Public Health Central Laboratory, LACEN, Salvador, BA, Brazil, 3 Molecular Epidemiology and Biostatistics Laboratory, Gonçalo Moniz Institute, FIOCRUZ-BA, Salvador, BA, Brazil, 4 Center for Integration of Data and Health Knowledge, CIDACS, Gonçalo Moniz Institute, FIOCRUZ-BA, Salvador, BA, Brazil, 5 Bahiana School of Medicine and Public Health, EBMSP, Salvador, BA, Brazil

* fernanda.grassi@fiocruz.br

**Data Availability Statement:** All relevant data are within the manuscript and its Supporting Information files.

## Abstract

Both Human T-lymphotropic virus type 1 (HTLV-1) and hepatitis C virus (HCV) are endemic in Brazil. In Salvador, the capital of the state of Bahia, 2% and 1.5% of the general population is infected with HTLV-1 or HCV. This study aimed to estimate the prevalence and the distribution of HTLV/HCV coinfection in Bahia. This cross-sectional study was conducted at the Central Laboratory of Public Health for the state of Bahia (LACEN-BA). All samples in the LACEN database submitted to serological testing for anti-HCV (chemiluminescence) and anti-HTLV-1/2 (chemiluminescence/ELISA and Western blot) from 2004 to 2013 were included. Infection rate was expressed as the number of infected individuals per 100,000 inhabitants in a given municipality; municipalities were grouped by microregion for further analysis. A total of 120,192 samples originating from 358 of the 417 municipalities in Bahia (85.8%) were evaluated. The overall HCV coinfection rate in HTLV-positive was 14.31% [2.8 (ranging from 0.4 to 8.0) per 100,000 inhabitants.] Twenty-one (5%) of the municipalities reported at least one case of HTLV/HCV coinfection. Most cases (87%) were concentrated in three microregions (Salvador: 79%, Ilhéus/Itabuna: 5%, Porto Seguro: 3%). Coinfection occurred more frequently in males (51%) with a mean age of 59 [(IQR): 46–59] years. HTLV/HCV coinfection in the state of Bahia was more frequently found among males living in the microregions of Salvador, Ilhéus/Itabuna and Porto Seguro, all of which are known to be endemic for HTLV infection.

## Introduction

Both human T-Lymphotropic virus (HTLV) and hepatitis C virus (HCV) are transmitted by parenteral exposure to contaminated blood or blood products [1–3]. In addition, HTLV can be transmitted sexually [4] and vertically from mother-to-child, predominantly through

**Funding:** This work was supported by the Coordination of Superior Level Staff Improvement-Brazil (CAPES) - Finance Code 001 and National Foundation for the Development of Private Higher Education (FUNADESP), grants 9600140 and 9600141. Maria Fernanda R. Grassi and Bernardo Galvão-Castro are research fellows of CNPq (process no. 304811/2017-3 and 311054/2014-5, respectively).

**Competing interests:** The authors have declared that no competing interests exist.

breastfeeding [5, 6]. HTLV type 1 (HTLV-1) is endemic in several parts of the world, with an estimated 5 to 10 million people harboring this virus [7]. In Brazil, the prevalence of HTLV-1 varies according to geographic location, with the North and Northeast regions being the most affected [8]. Recently, it was reported that HTLV-1 is widespread throughout the state of Bahia, with at least 130,000 individuals infected with this virus [9]. While HCV infection affects around 2.5% of the world's population (177.5 million adults), a population-based study in Brazil focusing on all 26 state capitals and the Federal District found an overall HCV sero-prevalence of 1.38% [10]; in Salvador, 1.5% of the general population is estimated to be infected with HCV [11].

In some regions endemic for HTLV infection, such as Asia and sub-Saharan Africa, the prevalence of HTLV/HCV coinfection in urban areas can reach up to 28% [12–15]. However, this coinfection has not been reported in Ethiopian rural areas [16]. In Europe, where the rate of HTLV infection in the general population is below 0.1% [7], although no reports of HTLV-1/HCV coinfection have been published to date [17, 18], HTLV-2/HCV coinfection was reportedly found in injecting drug users [19].

Brazil is endemic for both HTLV and HCV, and the presence of coinfection has been rarely reported in HCV patients undergoing treatment, as well as in blood donors, especially those in Southeastern Brazil. The prevalence of HTLV in individuals infected with HCV ranges from 5.3% in São Paulo [20] to 7.5% in Rio de Janeiro [21]. In addition, in blood donors, HCV was found in 35.9% of first-time HTLV-positive blood donors [22].

Contradictory clinical outcomes in the course of HTLV/HCV coinfection have been reported in Brazil and Japan. A better prognosis was described in coinfected Brazilian individuals who presented higher levels of Th1-type cytokines and CD4+ T lymphocytes, as well as lower hepatic fibrosis and alanine aminotransferase (ALT) [23–26]. Conversely, a Japanese study involving coinfected individuals described higher viral loads, a more rapid progression to hepatocellular carcinoma and a decreased response to treatment with interferon [15, 27–30]. In light of considerations regarding the influence of HTLV/HCV coinfection on the outcome of either infection and a lack of epidemiologic studies, notably in the Brazilian Northeast macroregion, the present study aimed to determine the rate of coinfection throughout the state of Bahia and map the geographical distribution of cases over a 10-year period.

## Materials and methods

### Ethics statement

The Institutional Review Board (IRB) for Human Research at the Gonçalo Moniz Institute of the Oswaldo Cruz Foundation (Salvador, Bahia, Brazil) provided ethical approval to conduct this study (CAAE number 22478813.7.0000.0040). In order to maintain patient information confidentiality, data were fully were anonymized so that the researchers do not have access to patient's individual information avoiding the need for verbal or written consent.

### Study area

The present study was carried out in the state of Bahia, Brazil, the fourth largest state in terms of population size (15,203,934 inhabitants), and the fifth-largest in terms of area: 564.722,611 km2. Bahia is comprised of 417 municipalities, which are grouped into 32 microregions and seven mesoregions according to economic and social similarities by the Brazilian National Institute of Geography and Statistics (IBGE) (http://www.ibge.gov.br).

## Study design

The present ecological retrospective study was using data obtained from the Central Laboratory of Public Health of Bahia (LACEN-BA), which is responsible for the laboratory analysis of infectious disease surveillance throughout the state. The population served by LACEN mainly consists of individuals who exhibit symptoms of infectious disease, pregnant women and individuals referred by blood blanks, the prison system or public health units distributed throughout the state of Bahia. Individuals were included if submitted to serological testing for both HTLV and HCV, either concomitantly or in isolation, from 2004 and 2013. Any indeterminate confirmatory test results were excluded from the present analysis.

## Laboratory testing

Serology for HTLV was performed at LACEN using the Murex HTLV-1/2 kit (DiaSorin SpA, Dartford, UK) from 2004 to 2008, the anti-HTLV-1/2 Sym Solution kit (Symbiosis Diagnostica LTDA, Leme, Brazil) from 2009 to 2010 and by microparticle CLIA chemiluminescence (Architect rHTLV-1/2, Abbott Diagnostics Division, Wiesbaden, Germany) from 2011 on. Confirmatory Western blotting (HTLV Blot 2.4, Genelabs Diagnostics, Singapore) was performed for all samples presenting seroreactivity. Serology for HCV was performed at LACEN by microparticle enzyme immunoassay (MEIA; AxSYM Anti-HCV Abbott Diagnostics Division, Illinois, USA) from 2004 to 2007, and thereafter by chemiluminescent microparticle immunoassay (Architect Anti-HCV, Abbott Diagnostics Division, Wiesbaden, Germany). With respect to molecular testing, RT-PCR (AMPLICOR MONITOR®, Roche Molecular Systems, Branchburg, NJ, USA) was employed in accordance with the manufacturer's specifications. Genotyping was performed by analyzing the highly conserved 5' untranslated region using the Linear Array Hepatitis C Virus Genotyping Test (LiPA—Line Probe Assay—Roche Diagnostics, USA), following the manufacturer's guidelines. This assay allows for the determination of six genotypes and subtypes (1a, 1b, 2, 2a, 2b, 3, 3a, 4, 4c, 5, 5a and 6).

## Data analysis

The SMART LAB laboratory management system was used to extract data from all serological HTLV and HCV tests performed throughout the study period, considering all individuals who were submitted to at least one HTLV and one HCV test. To avoid duplication, the most recent available serological results were considered. Each individual's unique registration number was considered as the key variable. The resulting database was validated using the R software package and analyzed with STATA v13.0. With regard to age, median and interquartile range (IQR) intervals were calculated and individuals were grouped accordingly. The prevalence of HTLV / HCV coinfection was determined by taking into account the presence of HTLV and HCV antibodies. The presence of anti-HTLV indicates infection, while the presence of anti-HCV might indicate exposure and / or infection. The rate of coinfection the primary outcome and was expressed as the number of individuals infected per 100,000 inhabitants. Digital maps detailing municipalities and microregions were obtained from the IBGE cartographic database in shapefile (.shp) format, and reformatted and analyzed using TerraView version 4.2 software freely provided by the National Institute for Space Research (www.dpi.inpe.br/terraview).

## Results

A total of 120,192 individuals were submitted to both anti-HCV and anti-HTLV 1/2 serology sometime from 2004 and 2013 (S1 Table). The median age of the studied population was 40 years [interquartile range (IQR): 25–41 years] and the female:male ratio was 7:1. The 861

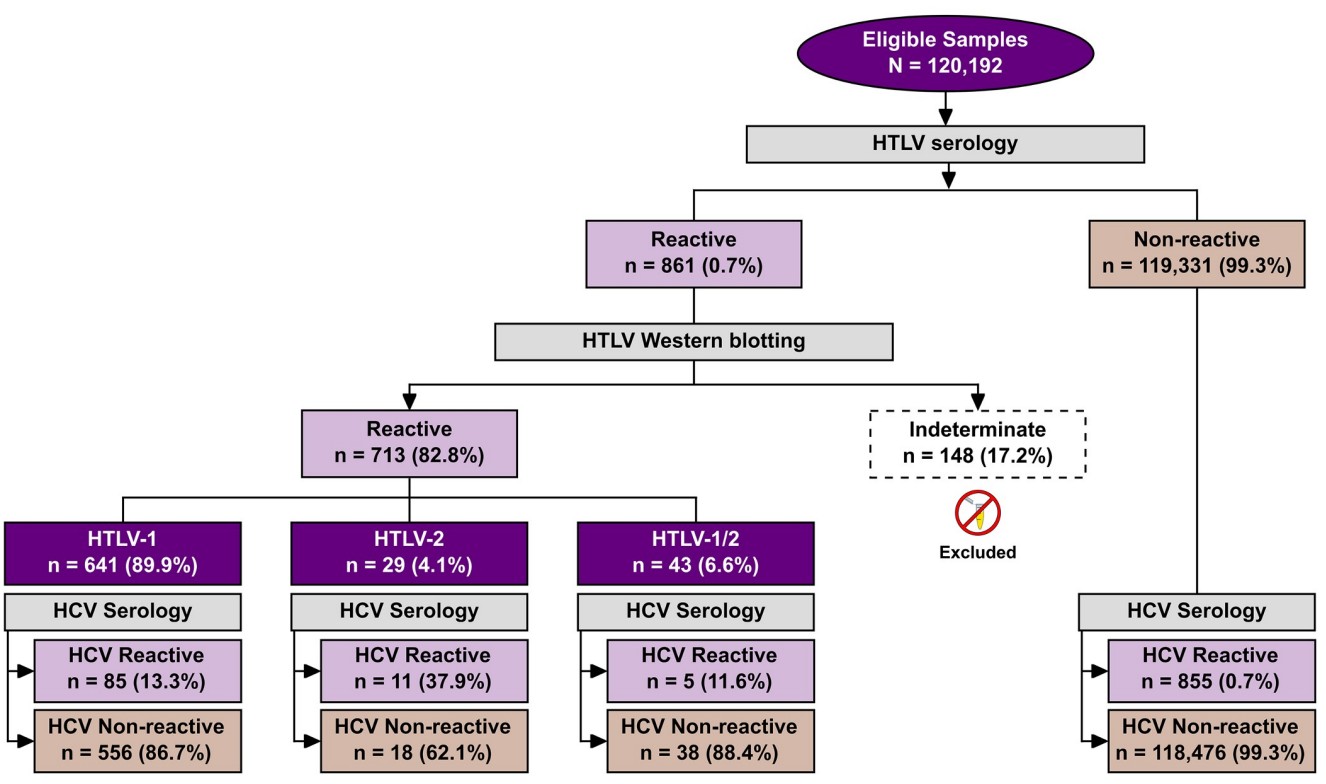

**Fig 1. Flow chart detailing the classification protocols employed in the studied population to determine HTLV and HCV infection status.**

HTLV-positive samples submitted to confirmatory Western blotting produced 713 [0.6%, 95% CI: 0.55–0.64, (713/120,192)] results with seroreactivity for HTLV: 641 (90%) were positive for HTLV-1, 29 (4.0%) for HTLV-2, and 43 (6%) were positive for both HTLV-1 and HTLV-2. The 148 (17.2%) indeterminate serologies were excluded from all further analysis (Fig 1).

HCV infection was detected in 14.2% (101/713; 95% CI: 11.9–17.1) of the HTLV-positive individuals, versus 0.72% (855/119,331; 95% CI: 0.67–0.77) of the HTLV-negative samples, p< 0.0001 (OR: 22.9; CI: 18.3–28.5) (Fig 1). With regard to coinfection, HCV infection was detected mainly among HTLV-2 (37.9%) infected individuals, followed by HTLV-1 (13.3%) and HTLV-1/2 (11.6%). The HTLV infection prevalence in the 120,192 individuals submitted to both anti-HCV and anti-HTLV-1/2 was substantially higher in women than in man subjects (75.6% vs. 24.4%, p < 0.001). The coinfection rate was more frequent in men (~30%, 52/174; p< 0.0001) compared to women (~9%, 49/534). The median age of cases de HTLV/HCV was 59 years [interquartile range (IQR): 46–59 years]. No age was registered for 16 coinfected individuals (Table 1). The genotype of HCV was identified in 31 out of 101 coinfected individuals. Gentotypes 1 (87%) and 3 (13%) were the most frequently identified.

Out of the 417 state municipalities, 358 (85.8%) sent samples to LACEN at some point during the study period. Information regarding sample origin was absent for 1.2% of the analyzed samples. The overall rate of HTLV/HCV coinfection was estimated at 2.8 per 100,000 inhabitants (range from 0.43 to 8.02 per 100,000 inhabitants). At least one case of HTLV or HCV infection was reported in 135 and 123 of the municipalities, respectively (Fig 2). Regarding the geographical distribution of coinfected HTLV/HCV cases, 21 (5.03%) municipalities presented at least one occurrence of coinfection. The majority of cases were concentrated in just three

**Table 1. Profile of HTLV/HCV coinfection in municipalities located in Bahia, Brazil from 2004 to 2013.**

| Municipality | Population* | Microregion | # performed tests (%Female) | # cases | Age | %Female | Rate** |
|---|---|---|---|---|---|---|---|
| Terra Nova | 12,467 | Catu | 224 (98) | 1 | 46 | 0 | 8.02 |
| Itagi | 14,084 | Jequié | 151 (87) | 1 | 48 | 100 | 7.10 |
| Biritinga | 14,129 | Serrinha | 580 (97) | 1 | 32 | 100 | 7.08 |
| Ipiaú | 43,169 | Ilhéus/Itabuna | 1,496 (91) | 2 | 57 | 0 | 4.63 |
| Itabela | 26,228 | Porto Seguro | 323 (93) | 1 | 47 | 0 | 3.81 |
| Camacan | 30,677 | Ilhéus/Itabuna | 647 (70) | 1 | 66 | 0 | 3.26 |
| Esplanada | 31,852 | Entre Rios | 517 (98) | 1 | 47 | 100 | 3.14 |
| Morro do Chapéu | 34,276 | Jacobina | 1,464 (92) | 1 | 27 | 100 | 2.92 |
| Vera Cruz | 35,951 | Salvador | 231 (94) | 1 | 56 | 100 | 2.78 |
| Senhor do Bonfim | 73,955 | Senhor do Bonfim | 1,016 (74) | 2 | 30 | 50 | 2.70 |
| Salvador | 2,920,679 | Salvador | 30,001 (81) | 78 | 54£ | 44.8 | 2.67 |
| São Sebastião do Passé | 40,972 | Catu | 140 (86) | 1 | 39 | 100 | 2.44 |
| Ipirá | 60,891 | Feira de Santana | 2,831 (93) | 1 | 50 | 0 | 1.64 |
| Brumado | 63,391 | Brumado | 276 (93) | 1 | 40 | 100 | 1.58 |
| Guanambi | 77,691 | Guanambi | 595 (89) | 1 | 51 | 0 | 1.29 |
| Santo Antônio de Jesus | 86,014 | Santo Antônio de Jesus | 1,644 (68) | 1 | 51 | 100 | 1.16 |
| Ilhéus | 219,927 | Ilhéus/Itabuna | 201 (56) | 2 | NA | 50 | 0.91 |
| Porto Seguro | 117,402 | Porto Seguro | 306 (86) | 1 | 60 | 100 | 0.85 |
| Teixeira de Freitas | 121,268 | Porto Seguro | 1,181 (85) | 1 | 63 | 100 | 0.82 |
| Lauro de Freitas | 147,661 | Salvador | 2,619 (87) | 1 | 52 | 100 | 0.68 |
| Juazeiro | 234,082 | Juazeiro | 93 (71) | 1 | 33 | 100 | 0.43 |
| | | TOTAL | | 101 | | | 2.8 |

Mean age (years)

*Mean population from 2008 to 2009 (www.ibge.gov.br).

** No. of cases per 100,000 inhabitants.

£ Mean age calculated from data available for 64 individuals.

microregions: Salvador (80/101 cases), Ilhéus/Itabuna (5/101 cases) and (Porto Seguro (3/101 cases) (Table 1 and Fig 2). In the Salvador microregion, 78 out of 80 cases of coinfection with both HTLV and HCV were concentrated the municipality of Salvador, followed by the municipalities of Ilhéus and Ipiaú (both located in Ilhéus/Itabuna microregion) with two identified cases each (Table 1).

## Discussion

The present study represents the first analysis of HTLV/HCV coinfection performed in the entire state of Bahia, which is considered endemic for both HTLV and HCV infections [11, 31]. A total of 14.2% of the HTLV-infected individuals investigated herein were found to be coinfected with HCV, resulting in an overall coinfection prevalence of 2.8/100,000 inhabitants. Although isolated infection with HTLV or HCV was disseminated throughout the state's microregions, the majority of HTLV/HCV coinfection cases clustered in just three microregions: Salvador (79% of cases), Ilhéus/Itabuna (5% of cases) and Porto Seguro (3% of cases). Of note, the Salvador microregion, which boasts a population of more than 3.4 million inhabitants, had the largest absolute number of coinfected individuals. This result was expected, since population-based studies conducted in this city reported that around 2% and 1.5% of individuals are infected with HTLV-1 and HCV, respectively [11, 31]. Salvador and the neighboring municipalities located around the Baía de Todos os Santos (Bay of All Saints) comprise

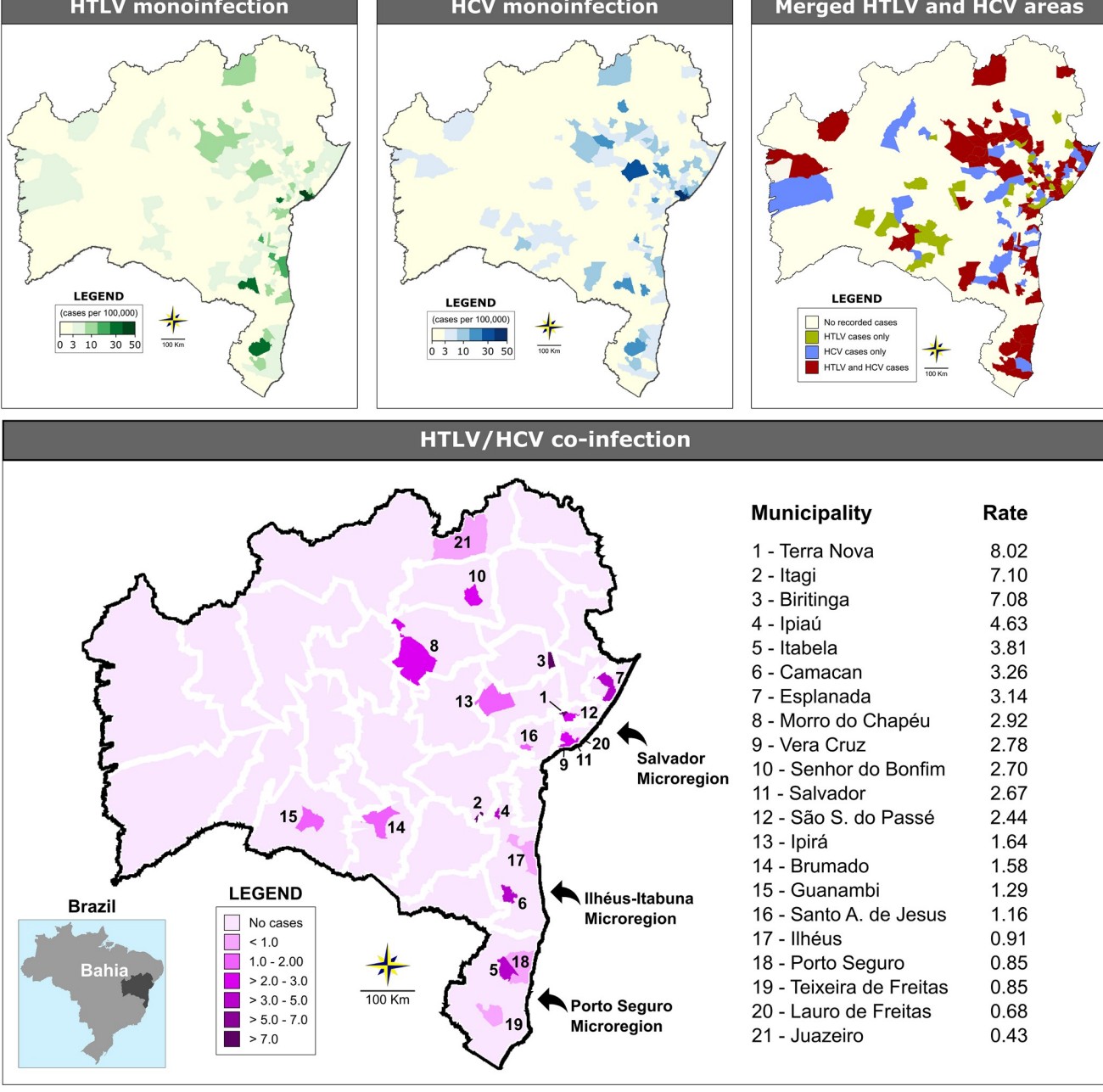

**Fig 2. Geographic distribution of HTLV and HCV cases among the municipalities in the state of Bahia from 2004 and 2013.** Microregions are delimited by white lines. The three microregions with the highest concentrations of coinfection cases are highlighted by arrows.

a region that was historically heavily engaged in slave trading from the 16th to 19th centuries. Nowadays, the local population is mainly of African descent. The economy is based on commercial, service, and industrial activities. In the Ilhéus-Itabuna and Porto Seguro microregions, both located in the southernmost region of the state, tourism and commercial activities are essential to the local economy. Recently, these two microregions were highlighted as significant foci of HTLV-1 infection [9].

The present study sought to determine the prevalence of HCV infection in a sample of HTLV-infected individuals. In Brazil, epidemiological data on the rate of HTLV/HCV coinfection remains scarce. Only three studies that estimated the prevalence of HTLV/HCV coinfection were recovered in the scientific literature: one in the city of São Paulo and one in the city of Rio de Janeiro, both studies evaluated the coinfection of HTLV / HCV in individuals undergoing HCV treatment [20, 21]. The third study was performed in the city of Ribeirão Preto (São Paulo) and evaluated for 11 years the seroprevalence of HTLV and co-infections with other sexually transmitted diseases such as HIV, syphilis and HBV [22]. In the first two studies, the denominator was the number of HCV cases, in the last one the denominator was the number of HTLV cases. Our study is similar to studies conducted in Japan, in which the prevalence of HTLV / HCV coinfection was estimated in HTLV endemic areas [12, 15] and in donor donors blood from Ribeirão Preto [22]. In contrast to our analysis, previous studies have evaluated HTLV infection in individuals with hepatitis C who received treatment [20, 21]. We found that HTLV-positive individuals presented a significantly higher prevalence for anti-HCV and also had a higher probability of acquiring hepatitis C (OR: 22.9) compared to HTLV-negative individuals. Similar results were obtained on Iki Island in Japan, an endemic region for both viruses [12]. However, this difference was not observed in a study conducted in two southern Japanese villages, where HTLV and HCV are endemic [15].

Throughout the study period, the testing methodology for both HTLV and HCV was updated from ELISA to chemiluminescence assays. Nonetheless, both tests offer similarly high sensitivity and specificity. Recent studies evaluating the performance of these methods indicate improved accuracy using chemiluminescence, both in HTLV [32] and HCV serology [33], i.e. both can be effectively used for serological screening in the laboratory diagnosis of HTLV and HCV. Chemiluminescence is mainly used by the laboratories that analyze a large number of samples, as this technique reduces time needed to produce results and minimizes errors in analysis.

In the present study, males over 50 years were predominantly found to be coinfected with HTLV and HCV, which is consistent with reports from other studies [20, 34, 35]. Moreover, it is known that the prevalence of HTLV-1 infection increases with age [31]. The advanced age of coinfected individuals was expected, since both HTLV and HCV have a long asymptomatic course, and the diagnosis of these infections often only occurs when chronic complications are present. In addition, since screening for both viruses only became mandatory at blood banks beginning in 1993, prior to this period, poor sterilization in the reuse of glass syringes was frequent, which contributed to increased transmission of HCV [36], and possibly also HTLV. Although the present study did not attempt to determine the routes of infection in co-infected individuals, it is possible that the sharing of needles and syringes was responsible for infection in these cases [10, 20, 34, 37, 38]. In HCV, which is transmitted through exposure to blood, particularly by transfusion, organ transplantation of infected individuals [39], injected drug use is considered the leading risk factor for virus acquisition [10, 40]. In contrast, HTLV-1 infection occurs more frequently by sexual route and breastfeeding, both uncommon in HCV transmission [41].

In the context of HTLV/HCV coinfection, the present study found a higher proportion of HTLV-2 in coinfected individuals. HTLV-2 infection has been mostly reported among injecting drug users in urban areas [8] and in indigenous populations in northern Brazil [42]. Achieving an accurate discrimination between HTLV-1 and HTLV-2 may have prognostic value in the outcome of HCV [20]. Some clinical complications, such as increased HCV load, hepatocyte damage, hepatocarcinogenesis, high levels of alanine aminotransferase (ALT) and increases in the spontaneous production of IL-1, IL-2 and IFN-γ, have been more frequently reported in individuals coinfected with HTLV-1/HCV than those with HTLV-2/HCV [23, 25, 26, 29, 34, 40, 43].

Regarding molecular testing, out of 101 HTLV/HCV-coinfected individuals included, only 40 records contained information regarding molecular investigation (RNA-HCV), and 31 of these were submitted to HCV genotyping. The Brazilian Ministry of Health protocol recommends genotyping for patients who are initiating treatment, as well as those undergoing treatment who present resistance to antivirals [44]. With respect to the profile of HCV circulating genotypes, the present study detected the presence of Genotypes 1 and 3, in 31 coinfected individuals. The most prevalent genotype was type 1 (83%). Accordingly, the HCV genotypes 1 and 3 are the most prevalent worldwide, accounting for about 46.2 and 30.1% of infections, respectively [45]. However, a low number of HCV genotype samples were tested, possibly leading to a misinterpretation of results, since the HCV clearance rate may vary in HTLV infected persons and by genotype.

The main limitation of the present study was the use of non-random sampling, which resulted in a predominance of females. In the state of Bahia, serological HTLV screening for pregnant women is compulsory since 2011, which could thusly contribute to a higher proportion of women in the samples analyzed. However, the robust representation of cases reported throughout the state's municipalities, as well as the absolute number of individuals analyzed, provide an overview of the circulation of HTLV and HCV in the state. Another limitation is using anti-HCV serology instead HCV-RNA measurement to determine the prevalence of infection. Different of HTLV, infection by HCV may be spontaneously cleared and treated. However, anti-HCV is the most common marker of hepatitis C infection, used to estimate its prevalence globally. In Brazil, it is one of the markers used to report cases of HCV infection in the SINAN [44]. Other studies previously conducted to estimate the prevalence of hepatitis C infection in Brazil and worldwide have also used anti-HCV for this purpose [8, 10, 46]. In the present study, we had information on RNA-HCV data for 40 HTLV/HCV-coinfected individuals. Of these, 32 had detectable viral load.

In conclusion, the present study evaluated a large number of individuals in Bahia, considered one of the states in Brazil most affected by HTLV infection. We found that at least 14% of the individuals infected with HTLV also harbor HCV. Coinfection was concentrated in males who resided in the microregions of Salvador, Ilhéus-Itabuna and Porto Seguro, all considered hotspots for HTLV infection, despite HTLV and HCV being widespread throughout the state. It is our hope that these findings will provide support for the implementation of preventive measures against the spread of these viruses, especially in areas where higher rates of HTLV/HCV coinfection were described. Moreover, as the presence of HTLV can negatively impact the course of HCV infection, the surveillance of active cases should be a priority in order to provide early treatment. The identification, control and prevention of the main risk factors associated with HTLV/HCV coinfection can lead to efficacious actions in a variety of epidemiological contexts specific to each affected region.

## Supporting information

**S1 Table. Anonymized data set comprised of HTLV and HCV tested individuals living in Bahia, Brazil from 2004 to 2013.**
(XLSX)

## Acknowledgments

We thank Andris K. Walter for providing English language revision and manuscript copyediting assistance.

## Author Contributions

**Conceptualization:** Felicidade Mota Pereira, Maria da Conceição Chagas de Almeida, Bernardo Galvão-Castro, Maria Fernanda Rios Grassi.

**Data curation:** Felicidade Mota Pereira, Maria da Conceição Chagas de Almeida, Roberto Perez Carreiro.

**Formal analysis:** Felicidade Mota Pereira, Fred Luciano Neves Santos, Roberto Perez Carreiro, Bernardo Galvão-Castro, Maria Fernanda Rios Grassi.

**Funding acquisition:** Bernardo Galvão-Castro, Maria Fernanda Rios Grassi.

**Investigation:** Felicidade Mota Pereira, Maria da Conceição Chagas de Almeida, Fred Luciano Neves Santos, Maria Fernanda Rios Grassi.

**Methodology:** Felicidade Mota Pereira, Fred Luciano Neves Santos, Maria Fernanda Rios Grassi.

**Project administration:** Maria Fernanda Rios Grassi.

**Resources:** Maria Fernanda Rios Grassi.

**Supervision:** Maria Fernanda Rios Grassi.

**Validation:** Felicidade Mota Pereira, Fred Luciano Neves Santos.

**Writing – original draft:** Felicidade Mota Pereira, Maria da Conceição Chagas de Almeida, Fred Luciano Neves Santos, Roberto Perez Carreiro, Bernardo Galvão-Castro, Maria Fernanda Rios Grassi.

**Writing – review & editing:** Felicidade Mota Pereira, Maria da Conceição Chagas de Almeida, Fred Luciano Neves Santos, Roberto Perez Carreiro, Bernardo Galvão-Castro, Maria Fernanda Rios Grassi.

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
