## [Decision Letter · Decision Letter 0]

16 Oct 2019

PONE-D-19-25685

Distribution of Human T-Lymphotropic Virus (HTLV) and Hepatitis C Co-infection in Bahia, Brazil,

PLOS ONE

Dear Dr. Santos,

Thank you for submitting your manuscript to PLOS ONE. After careful consideration, we feel that it has merit but does not fully meet PLOS ONE’s publication criteria as it currently stands. Therefore, we invite you to submit a revised version of the manuscript that addresses the points raised during the review process.

Both expert reviewers found several areas of the manuscript that require revision. Please provide a revised manuscript that addresses all comments and suggestions of both reviewers. For the HTLV and HCV serology, please include an explanation of the differences in assay sensitivity and specificity for the two different serological assays used for each virus and how that may have impacted the study results.

We would appreciate receiving your revised manuscript by Nov 30 2019 11:59PM. To enhance the reproducibility of your results, we recommend that if applicable you deposit your laboratory protocols in protocols.io, where a protocol can be assigned its own identifier (DOI) such that it can be cited independently in the future. For instructions see: http://journals.plos.org/plosone/s/submission-guidelines#loc-laboratory-protocols

We look forward to receiving your revised manuscript.

Kind regards,

William M. Switzer, MPH

Academic Editor

PLOS ONE

Journal Requirements:

2. In ethics statement in the manuscript and in the online submission form, please provide additional information about the patient records/samples used in your retrospective study. Specifically, please ensure that you have discussed whether all data/samples were fully anonymized before you accessed them and/or whether the IRB or ethics committee waived the requirement for informed consent. If patients provided informed written consent to have data/samples from their medical records used in research, please include this information.

'The funders had no role in study design, data collection and analysis, decision to

publish, or preparation of the manuscript.'

Please provide an amended Funding Statement that declares *all* the funding or sources of support received during this specific study (whether external or internal to your organization) as detailed online in our guide for authors at http://journals.plos.org/plosone/s/submit-now.  Please state what role the funders took in the study.  If any authors received a salary from any of your funders, please state which authors and which funder. If the funders had no role, please state: "The funders had no role in study design, data collection and analysis, decision to publish, or preparation of the manuscript."

Additional Editor Comments (if provided):

Reviewers' comments:

Reviewer's Responses to Questions

**Comments to the Author**

1. Is the manuscript technically sound, and do the data support the conclusions?

Reviewer #1: Partly

Reviewer #2: No

2. Has the statistical analysis been performed appropriately and rigorously? 

Reviewer #1: I Don't Know

Reviewer #2: No

3. Have the authors made all data underlying the findings in their manuscript fully available?

Reviewer #1: Yes

Reviewer #2: Yes

4. Is the manuscript presented in an intelligible fashion and written in standard English?

Reviewer #1: Yes

Reviewer #2: Yes

5. Review Comments to the Author

Reviewer #1: PONE-D-19-25685

Distribution of Human T-Lymphotropic Virus (HTLV) and Hepatitis C Co-infection in Bahia, Brazil

The manuscript contains novel information about HTLV/HCV coinfection and prevalence in Bahia and is based on very large cohort, covering approx. 0.8% of the state’s population.

Comments:

1. Please, include in the introduction why is the co-infection rate of those two agents important -clinical outcomes, treatment strategies, epidemiology, or not known at all? There is some information about this in the discussion, but it needs to be in the introduction to better justify the undertaking of the work. Lines 71-72 would be a good place to include such text.

2. Please, include in the description of the study design what was the no-random sampling bias that lead to testing predominantly females.

3. It is not clear from the data shown in the tables and figures why is co-infection more frequent in males at median age of 59? Please, demonstrate this conclusion with the specific results and support it with significance values.

4. Please, mention the genotyping method for HCV in the methods and explain why so few (only 30%) were successfully genotyped.

Reviewer #2: This manuscript aims to report on the prevalence of co-infection with both HCV and HTLV infection in the state of Bahia, Brazil, an area where HTLV is highly prevalent. The authors use a valuable data source from a central lab reporting source and analyses are generally well described and conducted. However, there are numerous issues that need to be addressed that would make this article have more public health and clinical impact. Why is mapping this important?

1) Throughout the manuscript the authors present different estimates of prevalence of co-infection: (a) in the general population; (b) prevalence of HTLV given HCV infection (i.e. the denominator is HCV infected) and (c) prevalence of HCV given HTLV infection. These all have different denominators, interpretations and epidemiological significance. They cannot be compared, yet the authors do this. For example, the paragraph that starts on line 66 this - what are the authors comparing and what is it they are focused on? is it overall HCV/HTLV in the whole population? or one given the other, and if so which? This needs to be reconciled throughout the manuscript for studies cited and for the results. Then the conclusions can be better articulated and interpreted. Note that the Discussion also conflated these different approaches and the interpretation is flawed as a result. (eg, the paragraph starting on line 177: we don't know what the denominator is for the Sao Paulo study and then this is compared to Japan where author present the prevalence of HCV given HTLV infection).

2) The authors have not presented the main rationale for the study - why is this analysis important to public health and or clinical practice? What gap does this analysis fill and how should it be used to improve health or change policy?

3) It would be very very good to know how the sample of people who were tested for BOTH viruses (which is the sample correct?) compare to the general population? it is hard to interpret this from what is presented. The authors state that because it is a large sample and it is noted that females are over-represented. I am assuming this is due to prenatal testing, but the authors should explain why this is. Several areas are 100% female samples. Then this needs to be better addressed in the limitations. The manuscript needs a table with demographics and primary outcome (HCV+HTLV exposure)

4) A major limitation is that the authors are presenting prevalence of EXPOSURE to HCV - as they only present anti-HCV results. Unlike HTLV, HCV infection can be spontaneously cleared and treated. Thus this estimate does not reflect prevalence of infection, but prevalence of exposure. If authors had HCV RNA results this would be better to use for the prevalence of co-infection estimate.

5) The low number of genotyped samples presents large interpretation issues: one because obviously only RNA positive samples can be genotyped, and there are potential biases resulting from (i) differential clearance rates of HCV found in HTLV-HCV infected persons, and (ii) potential differential clearance by genotype. This needs to be addressed in limitations.

Minor comments:

1) Introduction: (a) line 62: specify that this is one study conducted in rural Ethiopia. If possible let reader know if urban Ethiopia has higher co-prevalence; (b) line 65 - the conclusion of the study in Spain regarding people who inject drugs; - did this study have any data on country of origin? in particular African origin?

2) If you don't have HCV RNA results, then the manuscript needs to specify that this is HCV exposure and not HCV infection.

3) The Discussion needs to directly address how this data contributes to practice and or policy.

6. PLOS authors have the option to publish the peer review history of their article (what does this mean?). If published, this will include your full peer review and any attached files.

Reviewer #1: No

Reviewer #2: No

---

## [Author Response · Author response to Decision Letter 0]

16 Mar 2020

Dear Editor,

The response to the questions are below (PONE-D-19-25685). Additionally, we have included statements in the Ethics statement, Funding and Competing interests sections, as requested by Plos One Staff:

Ethics statement

Before: The Institutional Review Board (IRB) for Human Research at the Gonçalo Moniz Institute of the Oswaldo Cruz Foundation (Salvador, Bahia, Brazil) provided ethical approval to conduct this study (CAAE number 22478813.7.0000.0040). 

After (line 84-88): The Institutional Review Board (IRB) for Human Research at the Gonçalo Moniz Institute of the Oswaldo Cruz Foundation (Salvador, Bahia, Brazil) provided ethical approval to conduct this study (CAAE number 22478813.7.0000.0040). In order to maintain patient information confidentiality, data were fully were anonymized so that the researchers do not have access to patient’s individual information avoiding the need for verbal or written consent.

Funding

Before: No text. 

After (line 285-287): “The funders had no role in study design, data collection and analysis, decision to publish, or preparation of the manuscript.”

Competing interests

Before: No text. 

After (line 288-290): “The authors have declared that no competing interests exist.”

Editor: PONE-D-19-25685

Question 1 - Editor. Both expert reviewers found several areas of the manuscript that require revision. Please provide a revised manuscript that addresses all comments and suggestions of both reviewers. For the HTLV and HCV serology, please include an explanation of the differences in assay sensitivity and specificity for the two different serological assays used for each virus and how that may have impacted the study results.

Reply: We have included into the discussion section of the manuscript the following statement:

Before: No text.

After (line 216-223): “Throughout the study period, the testing methodology for both HTLV and HCV was updated from ELISA to chemiluminescence assays. Nonetheless, both tests offer similarly high sensitivity and specificity. Recent studies evaluating the performance of these methods indicate improved accuracy using chemiluminescence, both in HTLV [32] and HCV serology [33], i.e. both can be effectively used for serological screening in the laboratory diagnosis of HTLV and HCV. Chemiluminescence is mainly used by the laboratories that analyze a large number of samples, as this technique reduces time needed to produce results and minimizes errors in analysis.”

New references have been added to support the statements above:

(line 405-408): 32. Brito V da S, Santos FLN, Gonçalves NLS, Araujo TH, Nascimento DSV, Pereira FM, et al. Performance of commercially available serological screening tests for human T-cell lymphotropic virus infection in Brazil. J Clin Microbiol. 2018; pii: e00961-18. https://doi:10.1128/JCM.00961-18 PMID: 30232131.

(line 409-412): 33. Naz A, Mukry SN, Naseer I, Shamsi TS. Evaluation of efficacy of serological methods for detection of HCV infection in blood donors: A single centre experience. Pakistan J Med Sci. 2018; 34: 1204-1208. https://doi:10.12669/pjms.345.15707 PMID: 30344577.

Reviewer #1: PONE-D-19-25685

Question 1 - Reviewer 1. Please, include in the introduction why is the co-infection rate of those two agents important -clinical outcomes, treatment strategies, epidemiology, or not known at all? There is some information about this in the discussion, but it needs to be in the introduction to better justify the undertaking of the work. Lines 71-72 would be a good place to include such text.

Reply: We concur entirely with the reviewer with respect to the inclusion of further information regarding clinical outcomes, treatment strategies and the epidemiology of co-infection between HTLV and HCV agents. We have modified this section of the introduction to incorporate the reviewer’s suggestion.

Before: “Brazil is endemic for both HTLV and HCV, and the presence of coinfection has been reported in several populations, especially those in Southeastern Brazil. The prevalence of HTLV in individuals infected with HCV ranges from 5.3% in São Paulo (20) to 7.5% in Rio de Janeiro (21). In addition, in blood donors, HCV was found in 35.9% of first-time HTLV-positive blood donors (22) and 1.5% in HTLV-positive men who have sex with men (23).

In light of considerations regarding the influence of HTLV/HCV coinfection on the outcome of either infection, the present study aimed to determine the rate of coinfection throughout the state of Bahia and map the geographical distribution of cases over a 10-year period.”

After (line 66-80): “…Brazil is endemic for both HTLV and HCV, and the presence of coinfection has been rarely reported in HCV patients undergoing treatment, as well as in blood donors, especially those in Southeastern Brazil. The prevalence of HTLV in individuals infected with HCV ranges from 5.3% in São Paulo [20] to 7.5% in Rio de Janeiro [21]. In addition, in blood donors, HCV was found in 35.9% of first-time HTLV-positive blood donors [22]. 

Contradictory clinical outcomes in the course of HTLV/HCV coinfection have been reported in Brazil and Japan. A better prognosis was described in coinfected Brazilian individuals who presented higher levels of Th1-type cytokines and CD4+ T lymphocytes, as well as lower hepatic fibrosis and alanine aminotransferase (ALT) [23–26]. Conversely, a Japanese study involving coinfected individuals described higher viral loads, a more rapid progression to hepatocellular carcinoma and a decreased response to treatment with interferon [15,27–30]. In light of considerations regarding the influence of HTLV/HCV coinfection on the outcome of either infection and a lack of epidemiologic studies, notably in the Brazilian Northeast macroregion, the present study aimed to determine the rate of coinfection throughout the state of Bahia and map the geographical distribution of cases over a 10-year period.”

New references have been added to support the statements above:

 (line 372-375): 24. Abad-Fernández M, Moreno A, Dronda F, del Campo S, Quereda C, Casado JL, et al. Delayed liver fibrosis in HTLV-2-infected patients co-infected with HIV-1 and hepatitis C virus with suppressive antiretroviral therapy. AIDS. 2015; 29: 401-409. https://doi:10.1097/QAD.0000000000000555. PMID: 25565497.

 (line 385-388): 27. Kamihira S, Momita S, Ikeda S, Yamada Y, Sohda H, Atogami S, et al. Cohort study of hepatotropic virus and human T lymphotropic virus type-I infections in an area endemic for adult T cell leukemia. Jpn J Med. 1991; 30: 492-497. https://doi:10.2169/internalmedicine1962.30.492. PMID: 1665877.

(line 389-392): 28. Kishihara Y, Furusyo N, Kashiwagi K, Mitsutake A, Kashiwagi S, Hayashi J. Human T lymphotropic virus type 1 infection influences hepatitis C virus clearance. J Infect Dis. 2001; 184: 1114-1119. Epub 2001 Sep 21. https://doi:10.1086/323890. PMID: 11598833.

 (line 397-400): 30. Tokunaga M, Uto H, Oda K, Tokunaga M, Mawatari S, Kumagai K, et al. Influence of human T-lymphotropic virus type 1 coinfection on the development of hepatocellular carcinoma in patients with hepatitis C virus infection. J Gastroenterol. 2014; 49: 1567-1577. Epub 2014 Jan 25. https://doi:10.1007/s00535-013-0928-5 PMID: 24463696.

Question 2 - Reviewer 1. Please, include in the description of the study design what was the no-random sampling bias that lead to testing predominantly females.

Reply: We have modified “Material and Methods” and “Discussion” sections in the manuscript to incorporate the reviewer’s suggestions

Material and Methods section:

Before:

“Study design The present retrospective study was performed in the state of Bahia, Brazil, the fourth largest state in terms of population size (15,203,934 inhabitants), and the fifth largest in terms of area: 564.722,611 km2. Bahia is comprised of 417 municipalities, which are grouped into 32 microregions and seven mesoregions according to economic and social similarities by the Brazilian National Institute of Geography and Statistics (IBGE) (http://www.ibge.gov.br). Data were obtained from the Central Laboratory of Public Health of Bahia (LACEN-BA), which is responsible for the laboratory analysis of infectious disease surveillance throughout the state. Individuals were included if submitted to serological testing for both HTLV and HCV, either concomitantly or in isolation, between 2004 to 2013. Any indeterminate confirmatory test results were excluded from the present analysis.”

After (lines 87-101): “Study area The study was carried out in the state of Bahia, Brazil, the fourth largest state in terms of population size (15,203,934 inhabitants), and the fifth-largest in terms of area: 564.722,611 km2. Bahia is comprised of 417 municipalities, which are grouped into 32 microregions and seven mesoregions according to economic and social similarities by the Brazilian National Institute of Geography and Statistics (IBGE) (http://www.ibge.gov.br).

Study design The present ecological retrospective study was performed using data obtained from the Central Laboratory of Public Health of Bahia (LACEN-BA), which is responsible for the laboratory analysis of infectious disease surveillance throughout the state. The population served by LACEN mainly consists of individuals who exhibit symptoms of infectious disease, pregnant women and individuals referred by blood blanks, the prison system or public health units distributed throughout the state of Bahia. Individuals were included if submitted to serological testing for both HTLV and HCV, either concomitantly or in isolation, between 2004 to 2013. Any indeterminate confirmatory test results were excluded from the present analysis. 

Discussion section:

Before: “The main limitation of the present study was the use of non-random sampling, which resulted in a predominance of females. However, the robust representation of cases reported throughout the state’s municipalities, as well as the absolute number of individuals analyzed, provide an overview of the circulation of HTLV and HCV in the state.”

After (line 258-2631): “The main limitation of the present study was the use of non-random sampling, which resulted in a predominance of females. In the state of Bahia, serological HTLV screening for pregnant women is compulsory since 2011, which could thusly contribute to a higher proportion of women in the samples analyzed. However, the robust representation of cases reported throughout the state’s municipalities, as well as the absolute number of individuals analyzed, provide an overview of the circulation of HTLV and HCV in the state.” 

Question 3 - Reviewer 1. It is not clear from the date shown in the tables and figures why is co-infection more frequent in males at middle age of 59? Please demonstrate this conclusion with the specific results and support it with significance values.

Reply: The reviewer raises a valid point. In order to clarify this data in the manuscript, we have included the median age instead of simply reporting frequency. It is important to note that our analysis was conducted using a convenience sampling (i.e. not probabilistic). 

Before: “With regard to coinfection, HCV infection was detected mainly among HTLV-2 (37.9%) infected individuals, followed by HTLV-1 (13.3%) and HTLV-1/2 (11.6%). Coinfection was more frequent among males (51%, 52/101), with a median age of 59 years [interquartile range (IQR): 46 - 59 years].”

After (line 146-152): “With regard to coinfection, HCV infection was detected mainly among HTLV-2 (37.9%) infected individuals, followed by HTLV-1 (13.3%) and HTLV-1/2 (11.6%). The HTLV infection prevalence in the 120,192 individuals submitted to both anti-HCV and anti-HTLV-1/2 was substantially higher in women than in man subjects (75.6% vs. 24.4%, p < 0.001). The coinfection rate was more frequent in men (~30%, 52/174; p< 0.0001) compared to women (~9%, 49/534). The median age of cases de HTLV/HCV was 59 years [interquartile range (IQR): 46 - 59 years].”

Question 4 - Reviewer 1. Please, mention the genotyping method for HCV in the methods and explain why so few (only 30%) were successfully genotyped.

Reply: We have included in the methods section the genotyping method for HCV:

Before: No text.

After (line 111-117): “With respect to molecular testing, RT-PCR (AMPLICOR MONITOR®, Roche Molecular Systems, Branchburg, NJ, USA) was employed in accordance with the manufacturer's specifications. Genotyping was performed by analyzing the highly conserved 5’ untranslated region using the Linear Array Hepatitis C Virus Genotyping Test (LiPA - Line Probe Assay - Roche Diagnostics, USA), following the manufacturer’s guidelines. This assay allows for the determination of six genotypes and subtypes (1a, 1b, 2, 2a, 2b, 3, 3a, 4, 4c, 5, 5a and 6).”

In fact, only 30% of samples were successfully genotyped since this is a retrospective evaluation that considers a 10-year period (2004-2013). The database was generated from the records of a laboratory surveillance service that were sometimes lacking pertinent data. We have included in the discussion section the following sentence to clarify the low rate of HCV genotyping:

Before: No text. 

After (line 247-251): “Regarding molecular testing, out of 101 HTLV/HCV-coinfected individuals included, only 40 records contained information regarding molecular investigation (RNA-HCV), and 31 of these were submitted to HCV genotyping. The Brazilian Ministry of Health protocol recommends genotyping for patients who are initiating treatment, as well as those undergoing treatment who present resistance to antivirals [44].”

 

Reviewer #2: PONE-D-19-25685

Question 1 - Reviewer 2. Throughout the manuscript the authors present different estimates of prevalence of co-infection: (a) in the general population; (b) prevalence of HTLV given HCV infection (i.e. the denominator is HCV infected) and (c) prevalence of HCV given HTLV infection. These all have different denominators, interpretations and epidemiological significance. They cannot be compared, yet the authors do this. For example, the paragraph that starts on line 66 this - what are the authors comparing and what is it they are focused on? is it overall HCV/HTLV in the whole population? or one given the other, and if so which? This needs to be reconciled throughout the manuscript for studies cited and for the results. Then the conclusions can be better articulated and interpreted. Note that the Discussion also conflated these different approaches and the interpretation is flawed as a result. (eg, the paragraph starting on line 177: we don't know what the denominator is for the Sao Paulo study and then this is compared to Japan where author present the prevalence of HCV given HTLV infection).

Reply: Epidemiological studies on the rate of HTLV/HCV coinfection are extremely rare in Brazil. We initially observed that both HCV and HTLV are widespread in the state of Bahia, particularly in the city of Salvador in which two population-based studies were conducted to estimate the prevalence of these viruses. Only three studies that estimated the prevalence of HTLV/HCV coinfection were recovered in the scientific literature: one in the city of São Paulo and one in the city of Rio de Janeiro, both studies evaluated the coinfection of HTLV / HCV in individuals undergoing HCV treatment (Caterino De-Araujo et al., 2018); (Maciel and Mello, 2015). The third study was performed in the city of Ribeirão Preto / SP and evaluated for 11 years the seroprevalence of HTLV and co-infections with other STDs such as HIV, syphilis and HBV (Pinto et al., 2012). In the first two studies, the denominator was the number of HCV cases, in the last one the denominator was the number of HTLV cases. Our study is similar to studies conducted in Japan, in which the prevalence of HTLV / HCV coinfection was estimated in HTLV endemic areas (Nakashima et al., 1994; Boschi-Pinto et al., 2000) and in donor donors blood from Ribeirão Preto / SP (Pinto et al., 2012). We have modified the manuscript to make it clearly.

Results section:

Before: HCV infection was detected in 14.2% (101/713; 95% CI: 11.9-17.1) of the HTLV-positive individuals, versus 0.72% (855/119,331; 95% CI: 0.67 - 0.77) of the HTLV-negative samples (Figure 1).

After (line 144-146): HCV infection was detected in 14.2% (101/713; 95% CI: 11.9-17.1) of the HTLV-positive individuals, versus 0.72% (855/119,331; 95% CI: 0.67 - 0.77) of the HTLV-negative samples, p< 0.0001 (OR: 22.9; CI: 18.3 – 28.5) (Figure 1).

Discussion section:

Before: “The frequency of HTLV/HCV coinfection in the present study was slightly higher than that reported in São Paulo (5.3%), which is where more than one-half of all HCV-infected cases are concentrated in Brazil (20,34). However, this frequency was lower than rates found in two areas endemic for HTLV infection in Japan, in which around 25% of HTLV-1-infected individuals were also HCV-positive (12, 15).”

After (line 198-215): “The present study sought to determine the prevalence of HCV infection in a sample of HTLV-infected individuals. In Brazil, epidemiological data on the rate of HTLV/HCV coinfection remains scarce. Only three studies that estimated the prevalence of HTLV/HCV coinfection were recovered in the scientific literature: one in the city of São Paulo and one in the city of Rio de Janeiro, both studies evaluated the coinfection of HTLV / HCV in individuals undergoing HCV treatment [20,21]. The third study was performed in the city of Ribeirão Preto (São Paulo) and evaluated for 11 years the seroprevalence of HTLV and co-infections with other sexually transmitted diseases such as HIV, syphilis and HBV [22]. In the first two studies, the denominator was the number of HCV cases, in the last one the denominator was the number of HTLV cases. Our study is similar to studies conducted in Japan, in which the prevalence of HTLV / HCV coinfection was estimated in HTLV endemic areas [12,15] and in donor donors blood from Ribeirão Preto [22]. In contrast to our analysis, previous studies have evaluated HTLV infection in individuals with hepatitis C who received treatment [20,21]. We found that HTLV-positive individuals presented a significantly higher prevalence for anti-HCV and also had a higher probability of acquiring hepatitis C (OR: 22.9) compared to HTLV-negative individuals. Similar results were obtained on Iki Island in Japan, an endemic region for both viruses [12]. However, this difference was not observed in a study conducted in two southern Japanese villages, where HTLV and HCV are endemic [15].”

We have modified the conclusion as following:

Before: “In conclusion, the present study found that at least 14% of the individuals infected with HTLV in Bahia also harbor HCV. Coinfection was concentrated in males who resided in the microregions of Salvador, Ilhéus-Itabuna and Porto Seguro, all hotspots for HTLV infection. Given the clinical importance of identifying HTLV/HCV coinfection, new studies are needed to characterize the clinical and epidemiological profile of these individuals, in order to contribute to the prevention of these infections.”

After (line 272-277): “In conclusion, the present study found that at least 14% of the individuals infected with HTLV in Bahia also harbor HCV. Coinfection was concentrated in males who resided in the microregions of Salvador, Ilhéus-Itabuna and Porto Seguro, all hotspots for HTLV infection, although HTLV and HCV are widespread in the state. It is our hope that these findings will provide support for the implementation of measures to prevent infections caused by these viruses, especially in areas where higher rates of HTLV/HCV were described.”

 

Question 2 - Reviewer 2. The authors have not presented the main rationale for the study - why is this analysis important to public health and or clinical practice? What gap does this analysis fill and how should it be used to improve health or change policy?

Reply: The reviewer is correct in pointing out these deficiencies. Accordingly, we have included information in the introduction that highlights the importance of establishing the prevalence of coinfection with respect to impacts on clinical outcomes, treatment strategies and the epidemiology of co-infection between HTLV and HCV agents. Moreover, considering that the literature lacks substantial data regarding the prevalence of coinfection between HCV and HTLV, regardless of the denominator under consideration, our results provide support for the implementation of measures designed at preventing infections caused by these viruses in vulnerable populations, such as in individuals infected with HTLV, which is more prevalent in the state of Bahia than other regions in Brazil. 

Before: “Brazil is endemic for both HTLV and HCV, and the presence of coinfection has been reported in several populations, especially those in Southeastern Brazil. The prevalence of HTLV in individuals infected with HCV ranges from 5.3% in São Paulo (20) to 7.5% in Rio de Janeiro (21). In addition, in blood donors, HCV was found in 35.9% of first-time HTLV-positive blood donors (22) and 1.5% in HTLV-positive men who have sex with men (23).

In light of considerations regarding the influence of HTLV/HCV coinfection on the outcome of either infection, the present study aimed to determine the rate of coinfection throughout the state of Bahia and map the geographical distribution of cases over a 10-year period.”

After (line 66-80): “…Brazil is endemic for both HTLV and HCV, and the presence of coinfection has been rarely reported in HCV patients undergoing treatment, as well as in blood donors, especially those in Southeastern Brazil. The prevalence of HTLV in individuals infected with HCV ranges from 5.3% in São Paulo [20] to 7.5% in Rio de Janeiro [21]. In addition, in blood donors, HCV was found in 35.9% of first-time HTLV-positive blood donors [22]. 

Contradictory clinical outcomes in the course of HTLV/HCV coinfection have been reported in Brazil and Japan. A better prognosis was described in coinfected Brazilian individuals who presented higher levels of Th1-type cytokines and CD4+ T lymphocytes, as well as lower hepatic fibrosis and alanine aminotransferase (ALT) [23–26]. Conversely, a Japanese study involving coinfected individuals described higher viral loads, a more rapid progression to hepatocellular carcinoma and a decreased response to treatment with interferon [15,27–30]. In light of considerations regarding the influence of HTLV/HCV coinfection on the outcome of either infection and a lack of epidemiologic studies, notably in the Brazilian Northeast macroregion, the present study aimed to determine the rate of coinfection throughout the state of Bahia and map the geographical distribution of cases over a 10-year period.”

New references have been added to support the statements above:

 (line 372-375): 24. Abad-Fernández M, Moreno A, Dronda F, del Campo S, Quereda C, Casado JL, et al. Delayed liver fibrosis in HTLV-2-infected patients co-infected with HIV-1 and hepatitis C virus with suppressive antiretroviral therapy. AIDS. 2015; 29: 401-409. https://doi:10.1097/QAD.0000000000000555. PMID: 25565497.

 (line 385-388): 27. Kamihira S, Momita S, Ikeda S, Yamada Y, Sohda H, Atogami S, et al. Cohort study of hepatotropic virus and human T lymphotropic virus type-I infections in an area endemic for adult T cell leukemia. Jpn J Med. 1991; 30: 492-497. https://doi:10.2169/internalmedicine1962.30.492. PMID: 1665877.

(line 389-392): 28. Kishihara Y, Furusyo N, Kashiwagi K, Mitsutake A, Kashiwagi S, Hayashi J. Human T lymphotropic virus type 1 infection influences hepatitis C virus clearance. J Infect Dis. 2001; 184: 1114-1119. Epub 2001 Sep 21. https://doi:10.1086/323890. PMID: 11598833.

 (line 397-400): 30. Tokunaga M, Uto H, Oda K, Tokunaga M, Mawatari S, Kumagai K, et al. Influence of human T-lymphotropic virus type 1 coinfection on the development of hepatocellular carcinoma in patients with hepatitis C virus infection. J Gastroenterol. 2014; 49: 1567-1577. Epub 2014 Jan 25. https://doi:10.1007/s00535-013-0928-5 PMID: 24463696.

Question 3 - Reviewer 2 It would be very very good to know how the sample of people who were tested for BOTH viruses (which is the sample correct?) compare to the general population? it is hard to interpret this from what is presented. The authors state that because it is a large sample and it is noted that females are over-represented. I am assuming this is due to prenatal testing, but the authors should explain why this is. Several areas are 100% female samples. Then this needs to be better addressed in the limitations. The manuscript needs a table with demographics and primary outcome (HCV+HTLV exposure).

Reply: The authors are sympathetic to the reviewer’s request, and would sincerely like to offer robust prevalence rates for these infections in the general population for comparison purposes. However, since the LACEN laboratory, which is responsible for conducting these types of diagnoses throughout the state, is truly the only resource available capable of providing prevalence estimates, we regret that truly unbiased data with respect to the general population is not currently available. However, 120,192 individuals were included in the 10-year period of study, all of whom were tested for both HTLV and HCV viruses. The samples analyzed came from individuals who presented symptoms of infectious diseases, pregnant women or referred by blood banks, prisons and other health units of the public health system. In the state of Bahia, laboratory diagnosis of both HTLV and HCV are part of prenatal examinations, which certainly played a role in the greater representation of women in the study population. Although our sample is not representative of the general population due to these characteristics, it is in some way representative of the state as a whole with regard to the diversity of the data considered, since samples from 85.8% (358/417) municipalities of the state of Bahia were evaluated (total estimated state population: 15,126,371 inhabitants). Table 1 now includes the number of tested individuals per e municipality and the proposition of male/female. 

Table 1 (lines 172-175)

Municipality Population* # performed tests 

(%Female) Microregion # cases Age %Female Rate**

Terra Nova 12,467 224 (98) Catu 1 46 0 8.02

Itagi 14,084 151 (87) Jequié 1 48 100 7.10

Biritinga 14,129 580 (97) Serrinha 1 32 100 7.08

Ipiaú 43,169 1,496 (91) Ilhéus/Itabuna 2 57 0 4.63

Itabela 26,228 323 (93) Porto Seguro 1 47 0 3.81

Camacan 30,677 647 (70) Ilhéus/Itabuna 1 66 0 3.26

Esplanada 31,852 517 (98) Entre Rios 1 47 100 3.14

Morro do Chapéu 34,276 1,464 (92) Jacobina 1 27 100 2.92

Vera Cruz 35,951 231 (94) Salvador 1 56 100 2.78

Senhor do Bonfim 73,955 1,016 (74) Senhor do Bonfim 2 30 50 2.70

Salvador 2,920,679 30,001 (81) Salvador 78 54£ 44.8 2.67

São Sebastião do Passé 40,972 140 (86) Catu 1 39 100 2.44

Ipirá 60,891 2,831 (93) Feira de Santana 1 50 0 1.64

Brumado 63,391 276 (93) Brumado 1 40 100 1.58

Guanambi 77,691 595 (89) Guanambi 1 51 0 1.29

Santo Antônio de Jesus 86,014 1,644 (68) Santo Antônio de Jesus 1 51 100 1.16

Ilhéus 219,927 201 (56) Ilhéus/Itabuna 2 NA 50 0.91

Porto Seguro 117,402 306 (86) Porto Seguro 1 60 100 0.85

Teixeira de Freitas 121,268 1,181 (85) Porto Seguro 1 63 100 0.82

Lauro de Freitas 147,661 2,619 (87) Salvador 1 52 100 0.68

Juazeiro 234,082 93 (71) Juazeiro 1 33 100 0.43

 TOTAL 101 2.8

Question 4 - Reviewer 2 A major limitation is that the authors are presenting prevalence of EXPOSURE to HCV - as they only present anti-HCV results. Unlike HTLV, HCV infection can be spontaneously cleared and treated. Thus this estimate does not reflect prevalence of infection, but prevalence of exposure. If authors had HCV RNA results this would be better to use for the prevalence of co-infection estimate.

Reply: Based on the cited reference below we respectfully request that the reviewer reconsider his/her criticism. Indeed, anti-HCV is the most common marker of hepatitis C infection, used to estimate its prevalence globally. In Brazil, it is one of the markers used to report cases of HCV infection in the SINAN (Brazil, 2019). Other studies previously conducted to estimate the prevalence of hepatitis C infection in Brazil and worldwide have also used anti-HCV for this purpose (Pereira et al., 2013; Catalan-Soares et al., 2014; Petruzziello et al., 2016). In the present study, we had information on RNAHCV data for 40 HTLV/HCV-coinfected individuals. Of these, 32 had detectable viral load. 

Ministério da Saúde do Brasil. Boletim Epidemiológico - Hepatites virais. Brasília: Ministério da Saúde; 2018. Available: http://www.aids.gov.br. 

Pereira LM, Martelli CM, Moreira RC, Merchan-Hamman E, Stein AT, Cardoso MR, et al. Prevalence and risk factors of Hepatitis C virus infection in Brazil, 2005 through 2009: a cross-sectional study. BMC Infect Dis. 2013; 13: 60. https://doi.org/10.1186/1471-2334-13-60 PMID: 23374914. 

Petruzziello A, Marigliano S, Loquercio G, Cacciapuoti C. Hepatitis C virus (HCV) genotypes distribution: an epidemiological up-date in Europe. Infect Agent Cancer. 2016; 11: 53. https://doi:10.1186/s13027-016-0099-0 PMID: 27752280.

Before: No text.

After (line 263-271): “Another limitation is using anti-HCV serology instead HCV-RNA measurement to determine the prevalence of infection. Different of HTLV, infection by HCV may be spontaneously cleared and treated. However, anti-HCV is the most common marker of hepatitis C infection, used to estimate its prevalence globally. In Brazil, it is one of the markers used to report cases of HCV infection in the SINAN [44]. Other studies previously conducted to estimate the prevalence of hepatitis C infection in Brazil and worldwide have also used anti-HCV for this purpose [8,10,46]. In the present study, we had information on RNA-HCV data for 40 HTLV/HCV-coinfected individuals. Of these, 32 had detectable viral load.” 

Question 5 - Reviewer 2 The low number of genotyped samples presents large interpretation issues: one because obviously only RNA positive samples can be genotyped, and there are potential biases resulting from (i) differential clearance rates of HCV found in HTLV-HCV infected persons, and (ii) potential differential clearance by genotype. This needs to be addressed in limitations.

Reply: The study is a retrospective evaluation of a 10-year period 2004-2013. The database originates from a laboratory surveillance service, where available data were evaluated. Regarding the molecular tests, we identified that of the 101 HTLV/HCV-coinfected individuals, there were records of molecular examinations (RNA-HCV) for 40 individuals, of which 31 had resulted from the HCV genotype. Genotyping according to the Ministry of Health protocol is recommended for patients on initiation of treatment or those in the course of treatment who are resistant to antivirals. We also believe that part of the lack of information is due to discontinuity of health care and death. However, a low number of HCV genotype samples were tested, possibly leading to a misinterpretation of results, since the HCV clearance rate may vary in HTLV infected persons and by genotype.

Material and Methods section:

Before: “No text.”

After (line 111-117): “With respect to molecular testing, RT-PCR (AMPLICOR MONITOR®, Roche Molecular Systems, Branchburg, NJ, USA) was employed in accordance with the manufacturer's specifications. Genotyping was performed by analyzing the highly conserved 5’ untranslated region using the Linear Array Hepatitis C Virus Genotyping Test (LiPA - Line Probe Assay - Roche Diagnostics, USA), following the manufacturer’s guidelines. This assay allows for the determination of six genotypes and subtypes (1a, 1b, 2, 2a, 2b, 3, 3a, 4, 4c, 5, 5a and 6).”

Discussion section:

Before: No text.

After (line 247-257): Regarding molecular testing, out of 101 HTLV/HCV-coinfected individuals included, only 40 records contained information regarding molecular investigation (RNA-HCV), and 31 of these were submitted to HCV genotyping. The Brazilian Ministry of Health protocol recommends genotyping for patients who are initiating treatment, as well as those undergoing treatment who present resistance to antivirals [44]. With respect to the profile of HCV circulating genotypes, the present study detected the presence of Genotypes 1 and 3, in 31 coinfected individuals. The most prevalent genotype was type 1 (83%). Accordingly, the HCV genotypes 1 and 3 are the most prevalent worldwide, accounting for about 46.2 and 30.1% of infections, respectively [45]. However, a low number of HCV genotype samples were tested, possibly leading to a misinterpretation of results, since the HCV clearance rate may vary in HTLV infected persons and by genotype.

Question 6 - Reviewer 2 Introduction: (a) line 62: specify that this is one study conducted in rural Ethiopia. If possible let reader know if urban Ethiopia has higher co-prevalence; 

Reply: We thank the reviewer for his/her observation and have accordingly added the suggested information

Before: “However, this coinfection has not been reported rural areas (16). In Europe, where….”

After (line 61-62): “However, this coinfection has not been reported in Ethiopian rural areas (16). In Europe, where….

Question 7 - Reviewer 2 Introduction: (b) line 65 - the conclusion of the study in Spain regarding people who inject drugs; - did this study have any data on country of origin? in particular African origin?

Reply: The study population were composed of 351 individuals living in Barcelona, 422 in Madrid and 188 in Seville. Of these, 10.3% were foreigners, but do not report the country of origin.

Question 8 - Reviewer 2 If you don't have HCV RNA results, then the manuscript needs to specify that this is HCV exposure and not HCV infection.

Reply: Based on the cited reference below we respectfully request that the reviewer reconsider his/her criticism. Indeed, anti-HCV is the most common marker of hepatitis C infection, used to estimate its prevalence globally. In Brazil, it is one of the markers used to report cases of HCV infection in the SINAN (Brazil, 2019). Other studies previously conducted to estimate the prevalence of hepatitis C infection in Brazil and worldwide have also used anti-HCV for this purpose (Pereira et al., 2013; Catalan-Soares et al., 2014; Petruzziello et al., 2016). In the present study, we had information on RNAHCV data for 40 HTLV/HCV-coinfected individuals. Of these, 32 had detectable viral load. 

Ministério da Saúde do Brasil. Boletim Epidemiológico - Hepatites virais. Brasília: Ministério da Saúde; 2018. Available: http://www.aids.gov.br. 

Pereira LM, Martelli CM, Moreira RC, Merchan-Hamman E, Stein AT, Cardoso MR, et al. Prevalence and risk factors of Hepatitis C virus infection in Brazil, 2005 through 2009: a cross-sectional study. BMC Infect Dis. 2013; 13: 60. https://doi.org/10.1186/1471-2334-13-60 PMID: 23374914. 

Petruzziello A, Marigliano S, Loquercio G, Cacciapuoti C. Hepatitis C virus (HCV) genotypes distribution: an epidemiological up-date in Europe. Infect Agent Cancer. 2016; 11: 53. https://doi:10.1186/s13027-016-0099-0 PMID: 27752280.

Question 9 - Reviewer 2 The Discussion needs to directly address how this data contributes to practice and or policy.

Reply: We concur entirely with the reviewer and we have included a statement in the Discussion section.

Before: “In conclusion, the present study found that at least 14% of the individuals infected with HTLV in Bahia also harbor HCV. Coinfection was concentrated in males who resided in the microregions of Salvador, Ilhéus-Itabuna and Porto Seguro, all hotspots for HTLV infection. Given the clinical importance of identifying HTLV/HCV coinfection, new studies are needed to characterize the clinical and epidemiological profile of these individuals, in order to contribute to the prevention of these infections.”

After (line 272-277): “In conclusion, the present study found that at least 14% of the individuals infected with HTLV in Bahia also harbor HCV. Coinfection was concentrated in males who resided in the microregions of Salvador, Ilhéus-Itabuna and Porto Seguro, all hotspots for HTLV infection, although HTLV and HCV are widespread in the state. It is our hope that these findings will provide support for the implementation of measures to prevent infections caused by these viruses, especially in areas where higher rates of HTLV/HCV were described.”

---

## [Decision Letter · Decision Letter 1]

21 Apr 2020

PONE-D-19-25685R1

Distribution of Human T-Lymphotropic Virus (HTLV) and Hepatitis C Co-infection in Bahia, Brazil,

PLOS ONE

Dear PhD Santos,

Thank you for submitting your manuscript to PLOS ONE. After careful consideration, we feel that it has merit but does not fully meet PLOS ONE’s publication criteria as it currently stands. Therefore, we invite you to submit a revised version of the manuscript that addresses the points raised during the review process.

Specifically, please address the suggestions and comments of the reviewer raised in the current revision.

We would appreciate receiving your revised manuscript by Jun 05 2020 11:59PM. To enhance the reproducibility of your results, we recommend that if applicable you deposit your laboratory protocols in protocols.io, where a protocol can be assigned its own identifier (DOI) such that it can be cited independently in the future. For instructions see: http://journals.plos.org/plosone/s/submission-guidelines#loc-laboratory-protocols

We look forward to receiving your revised manuscript.

Kind regards,

William M. Switzer, MPH

Academic Editor

PLOS ONE

Reviewers' comments:

Reviewer's Responses to Questions

**Comments to the Author**

1. If the authors have adequately addressed your comments raised in a previous round of review and you feel that this manuscript is now acceptable for publication, you may indicate that here to bypass the “Comments to the Author” section, enter your conflict of interest statement in the “Confidential to Editor” section, and submit your "Accept" recommendation.

Reviewer #1: All comments have been addressed

Reviewer #2: (No Response)

2. Is the manuscript technically sound, and do the data support the conclusions?

Reviewer #1: Yes

Reviewer #2: Partly

3. Has the statistical analysis been performed appropriately and rigorously? 

Reviewer #1: Yes

Reviewer #2: Yes

4. Have the authors made all data underlying the findings in their manuscript fully available?

Reviewer #1: Yes

Reviewer #2: Yes

5. Is the manuscript presented in an intelligible fashion and written in standard English?

Reviewer #1: Yes

Reviewer #2: Yes

6. Review Comments to the Author

Reviewer #1: The authors have made a very diligent and thorough effort to address the comments and suggestions provided by the reviewers. As a result the quality of the manuscript has improved significantly.

Reviewer #2: The authors have mostly addressed my concerns and critiques in the revised submission, and I commend their attention to these. I have the following minor issues that I think need addressing to make the manuscript publishable.

1) The authors need to be explicit in the Methods section that this study used data from anti-HCV test results indicating exposure to HCV and that they are using this as a proxy for infection in their co-infection estimates. While they correctly noted in the limitations that clearance would result in fewer infections that estimated, this could be more explicitly stated in methods. Contrary to what they note, anti-HCV is not used globally to indicate infection, it is used to indicate exposure and previous infection. Many current systematic reviews use or estimate infection based on RNA results, or impute it based on expected clearance rates. Also as more people are treated, we can expect anti-HCV to be an even larger over-estimate of infection.

2) Lines 127-128: age) . authors say that they 'adjusted' the estimates. Adjusted for what? if its not adjusted, for example for age, then just say the rate per 100,000 and remove the "adjusted" . Also this sentence implies that your primary outcome is HCV and HTLV- co-infection among all tests reviewed. Not only HCV in those with HTLV. Is this correct?

3) The final conclusion is still weak: how can this data inform prevention? just hoping it will is not enough - what are ways that this paper can be used beyond their descriptive contribution?

7. PLOS authors have the option to publish the peer review history of their article (what does this mean?). If published, this will include your full peer review and any attached files.

Reviewer #1: No

Reviewer #2: No

---

## [Author Response · Author response to Decision Letter 1]

3 Jun 2020

Question 1 - Reviewer #2: The authors need to be explicit in the Methods section that this study used data from anti-HCV test results indicating exposure to HCV and that they are using this as a proxy for infection in their co-infection estimates. While they correctly noted in the limitations that clearance would result in fewer infections that estimated, this could be more explicitly stated in methods. Contrary to what they note, anti-HCV is not used globally to indicate infection, it is used to indicate exposure and previous infection. Many current systematic reviews use or estimate infection based on RNA results, or impute it based on expected clearance rates. Also as more people are treated, we can expect anti-HCV to be an even larger over-estimate of infection.

Reply: We have included the following sentence in the methods section:

Before: no information.

After (line 127-130): The prevalence of HTLV / HCV coinfection was determined by taking into account the presence of HTLV and HCV antibodies. The presence of anti-HTLV indicates infection, while the presence of anti-HCV might indicate exposure and / or infection. 

Question 2 - Reviewer #2: Lines 127-128: age) . authors say that they 'adjusted' the estimates. Adjusted for what? if its not adjusted, for example for age, then just say the rate per 100,000 and remove the "adjusted". 

Reply: We thank the reviewer by his/her observation. We have removed “adjusted”. The rate means the number of infected individual per 100.000 habitants.

Before: “The adjusted rate of coinfection was expressed as the number of individuals infected per 100,000 inhabitants.”

After (line 130-131): “The rate of coinfection was expressed as the number of individuals infected per 100,000 inhabitants.”

Question 3 - Reviewer #2: The final conclusion is still weak: how can this data inform prevention? Just hoping it will is not enough - what are ways that this paper can be used beyond their descriptive contribution?

 Reply: We have modified the conclusion as follows:

Before: “In conclusion, the present study found that at least 14% of the individuals infected with HTLV in Bahia also harbor HCV. Coinfection was concentrated in males who resided in the microregions of Salvador, Ilhéus-Itabuna and Porto Seguro, all hotspots for HTLV infection, although HTLV and HCV are widespread in the state. It is our hope that these findings will provide support for the implementation of measures to prevent infections caused by these viruses, especially in areas where higher rates of HTLV/HCV were described.”

After (line 276-287): “In conclusion, the present study evaluated a large number of individuals in Bahia, considered one of the states in Brazil most affected by HTLV infection. We found that at least 14% of the individuals infected with HTLV also harbor HCV. Coinfection was concentrated in males who resided in the microregions of Salvador, Ilhéus-Itabuna and Porto Seguro, all considered hotspots for HTLV infection, despite HTLV and HCV being widespread throughout the state. It is our hope that these findings will provide support for the implementation of preventive measures against the spread of these viruses, especially in areas where higher rates of HTLV/HCV coinfection were described. Moreover, as the presence of HTLV can negatively impact the course of HCV infection, the surveillance of active cases should be a priority in order to provide early treatment. The identification, control and prevention of the main risk factors associated with HTLV/HCV coinfection can lead to efficacious actions in a variety of epidemiological contexts specific to each affected region.”

---

## [Editor Report · Decision Letter 2]

4 Jun 2020

PONE-D-19-25685R2

Distribution of Human T-Lymphotropic Virus (HTLV) and Hepatitis C Co-infection in Bahia, Brazil,

PLOS ONE

Dear Dr. Santos,

Thank you for submitting your manuscript to PLOS ONE. After careful consideration, we feel that it has merit but still does not fully meet PLOS ONE’s publication criteria as it currently stands. Therefore, we invite you to submit a revised version of the manuscript that addresses the minor points raised by one reviewer during the review process.

Please revise the manuscript in accordance with the additional minor recommendations of Dr. Page.

We look forward to receiving your revised manuscript.

Kind regards,

William M. Switzer, MPH

Academic Editor

PLOS ONE

Additional Editor Comments (if provided):

Please revise the manuscript in accordance with the additional minor recommendations of Dr. Page.

---

## [Author Response · Author response to Decision Letter 2]

15 Jun 2020

Dear Editor,

Thank you very much for analyzing our manuscript. The replies to Editor’s comments are below:

 Reviewer #2: The authors have mostly addressed my concerns and critiques in the revised submission, and I commend their attention to these. I have the following minor issues that I think need addressing to make the manuscript publishable.

Question 1 - Reviewer #2: The authors need to be explicit in the Methods section that this study used data from anti-HCV test results indicating exposure to HCV and that they are using this as a proxy for infection in their co-infection estimates. While they correctly noted in the limitations that clearance would result in fewer infections that estimated, this could be more explicitly stated in methods. Contrary to what they note, anti-HCV is not used globally to indicate infection, it is used to indicate exposure and previous infection. Many current systematic reviews use or estimate infection based on RNA results, or impute it based on expected clearance rates. Also as more people are treated, we can expect anti-HCV to be an even larger over-estimate of infection.

Reply: We have included the following sentence in the methods section:

Before: no information.

After (line 127-130): The prevalence of HTLV / HCV coinfection was determined by taking into account the presence of HTLV and HCV antibodies. The presence of anti-HTLV indicates infection, while the presence of anti-HCV might indicate exposure and / or infection. 

Question 2 - Reviewer #2: Lines 127-128: age) . authors say that they 'adjusted' the estimates. Adjusted for what? if its not adjusted, for example for age, then just say the rate per 100,000 and remove the "adjusted". Also this sentence implies that your primary outcome is HCV and HTLV- co-infection among all tests reviewed. Not only HCV in those with HTLV. Is this correct?

Reply: We thank the reviewer by his/her observation. We have removed “adjusted”. The rate means the number of infected individual per 100.000 habitants. The rate means the number of infected individual per 100.000 habitants and our primary outcome is HCV and HTLV coinfection.

Before: “The adjusted rate of coinfection was expressed as the number of individuals infected per 100,000 inhabitants.”

After (line 130-131): “The rate of coinfection was the primary outcome and was expressed as the number of individuals infected per 100,000 inhabitants.”

Question 3 - Reviewer #2: The final conclusion is still weak: how can this data inform prevention? Just hoping it will is not enough - what are ways that this paper can be used beyond their descriptive contribution?

 Reply: We have modified the conclusion as follows:

Before: “In conclusion, the present study found that at least 14% of the individuals infected with HTLV in Bahia also harbor HCV. Coinfection was concentrated in males who resided in the microregions of Salvador, Ilhéus-Itabuna and Porto Seguro, all hotspots for HTLV infection, although HTLV and HCV are widespread in the state. It is our hope that these findings will provide support for the implementation of measures to prevent infections caused by these viruses, especially in areas where higher rates of HTLV/HCV were described.”

After (line 277-288): “In conclusion, the present study evaluated a large number of individuals in Bahia, considered one of the states in Brazil most affected by HTLV infection. We found that at least 14% of the individuals infected with HTLV also harbor HCV. Coinfection was concentrated in males who resided in the microregions of Salvador, Ilhéus-Itabuna and Porto Seguro, all considered hotspots for HTLV infection, despite HTLV and HCV being widespread throughout the state. It is our hope that these findings will provide support for the implementation of preventive measures against the spread of these viruses, especially in areas where higher rates of HTLV/HCV coinfection were described. Moreover, as the presence of HTLV can negatively impact the course of HCV infection, the surveillance of active cases should be a priority in order to provide early treatment. The identification, control and prevention of the main risk factors associated with HTLV/HCV coinfection can lead to efficacious actions in a variety of epidemiological contexts specific to each affected region.”

---

## [Editor Report · Decision Letter 3]

30 Jun 2020

Distribution of Human T-Lymphotropic Virus (HTLV) and Hepatitis C Co-infection in Bahia, Brazil,

PONE-D-19-25685R3

Dear Dr. Santos,

We’re pleased to inform you that your manuscript has been judged scientifically suitable for publication and will be formally accepted for publication once it meets all outstanding technical requirements.

Kind regards,

William M. Switzer, MPH

Academic Editor

PLOS ONE
---

## [Editor Report · Acceptance letter]

8 Jul 2020

PONE-D-19-25685R3 

Distribution of Human T-Lymphotropic Virus (HTLV) and Hepatitis C Co-infection in Bahia, Brazil, 

Dear Dr. Santos:

I'm pleased to inform you that your manuscript has been deemed suitable for publication in PLOS ONE. Congratulations! Your manuscript is now with our production department. 

Kind regards, 

on behalf of

Mr. William M. Switzer 

Academic Editor

PLOS ONE